# Effect of practice exergames on the mood states and self-esteem of elementary school boys and girls during physical education classes: A cluster-randomized controlled natural experiment

**Alexandro Andrade**[1]*, **Whyllerton Mayron da Cruz**[1], **Clara Knierim Correia**[2], **Ana Luiza Goya Santos**[1], **Guilherme Guimarães Bevilacqua**[1]

1 Laboratory of Sport and Exercise Psychology, Santa Catarina State University, Florianópolis, Brazil,
2 Laboratory of Aquatic Biomechanics, Santa Catarina State University, Florianópolis, Brazil

* alexandro.andrade.phd@gmail.com

## Abstract

Reduced physical exercise can impact children's mental health. Use of active electronic games can help promote psychological health. Physical education (PE class) uses different resources, methods, and contents to promote student health. We investigated the effect of exergames on the mood and self-esteem of children and compare it that of with traditional PE classes. From a sample of 213 children (7–11 years old), 140 from 10 classes of the fourth and fifth grades of elementary school (59 boys, 81 girls; mean age 9.41±0.48 years), allocated to an experimental group (EG; n = 68; five clusters) and a PE group (PE; n = 72; five clusters), participated in this experimental controlled study. The EG practiced exergames during three 40-minute classes, and the PE group held three routine curricular PE classes. Brunel's Mood Scale and Rosenberg's Self-Esteem Scale were applied. Repeated measures ANOVA identified differences between sexes and groups. The main results of the EG demonstrated reduced tension in girls (p <0.05; ES: 0.62; 95% CI: 0.17–1.05). Regarding sex comparisons, anger was lower in girls (F: 4.57; p <0.05; ES: 0.61; 95% CI: 0.11–1.11) in the EG. Vigor was higher in girls in the EG than in those in the PE group (F: 5.46; p <0.05; ES: 0.56; 95% CI: 0.12–1.01). The main results of the PE group indicated increased self-esteem in boys (p <0.05; ES: 0.58; 95% CI: 0.08–1.07) and reduction of girls' mental confusion (p <0.05; ES 0.58; 95% CI 0.15–1.06). Thus, exergames impact boys' and girls' self-esteem and mood, as well as traditional physical education classes. Further study on exergames in schools is essential, with long-term effects on physical and mental health. Exergames bring interesting, varied content, technology, and innovation that can increase the attractiveness of physical education.

## Introduction

Physical inactivity is a topic of worldwide concern and has been increasing in many countries. The current reality of an increasingly inactive or sedentary populous results in health problems

**Data Availability Statement:** All data underlying the findings described fully available in the manuscript.

**Funding:** CKC: Higher Education Personnel Improvement Coordination (CAPES), Public notice (n. 03/2015). FAPESC (Research and Innovation Support Foundation of the State of Santa Catarina) for financial support through research (PAP 01/ 2016).

**Competing interests:** The authors have declared that no competing interests exist.

such as hypertension, heart issues, and psychological disorders [1–3]. In part, this is due to limited time and opportunities for physical activity, ease of Internet access, and increased time spent using smartphones, in front of the television, or in sedentary video games, mainly manifesting in children and teenagers [4,5].

Especially in children, the impact of early inclusion into digital electronic activities has sparked interest in research on learning, sleep, and the relationship between these electronic activities and their effect on well-being and mental health [6–9]. Despite the negative effects, it has recently become possible to observe the potential benefits that technology can have on mental health and the involvement of physical activity [10–13].

It is remarkable that physical activity habits are an important component of a healthy lifestyle and health promotion [14]. In this perspective, the time that children spend in school becomes a key factor to raise awareness and opportunity to practice physical activity. Thus, school physical education, as a curricular discipline, can contribute considerably to the increase in desired daily physical activity and aim to positively affect physical and psychological health. The classes, in addition to traditional content (e.g., games, sports, dancing, fighting) and the use of new methodologies, can make physical activity much more attractive to young people and children [15].

In this context, the use of cyberspace combined with traditional contents of physical education at school has been used as an educational tool. Some studies seek to identify the impact of technologies within physical education classes. In addition to being more attractive and motivating, cyberspace contributes to students' involvement in physical activities during classes [16–19].

In this sense, active videogames are considered game modalities that seem to be suitable for both boys and girls because they combine fun with the use of technology, contributing several physical and psychological benefits [20,21]. Known as exergames, this class of active games has recently taken a prominent role as technological devices designed to increase physical activity and promote physical and mental health [22,23]. In recent years, these types of games have demonstrated their ability to improve physical fitness and to reduce body mass index [24,25].

Recently, some experimental studies using exergames in non-school settings have shown positive effects on the health of children and adolescents [17,26–28]. Specifically, on effects on mental health, other studies have shown improvement in body image [29], psychological well-being [30–32], and self-efficacy [18]. The implications related to psychological health also indicate improvements in self-esteem [33,34] and mood states [16].

Despite the above, studies addressing active games in curriculum proposals and addressing mental health issues within the school environment are scarce [35]. Experimental studies using exergames outside the school environment have low ecological validity, and the few integrated studies performed as a physical education component were performed only in one session [22]. These situations reflect controversial results with low reproducibility. Randomized experiments performed with exergames in the school environment are still rare [19,21], making the results even more inconclusive [10].

Mood studies have been developed in the sports and education sciences, in general, proving moods as an important variable in performance, health, and education [22,36–38]. It is necessary to investigate the psychological effects of different innovative methods during physical education classes at school because when considering the diversity of resources and methods such as exergames, it can present results that are still little known or not fully understood. The chronic and acute effects of physical education practice have been demonstrated in several experimental studies [39–41]. However, there is a lack of experiments with exergames and children in schools. We are aware that the practice of physical education has acute effects on physical and mental health [20,28,42]. It is important to know whether exergames also

significantly alter the mental health of children. Acute effects on mood and self-esteem, for example, can have a positive impact on many aspects of a child's life with consequences for school performance.

Our study investigated whether exergames influenced the mood and self-esteem of children at school, whether there would be an acute effect of this practice, and whether there would be differences between the sexes and between groups. Therefore, the objective of this study was to determine if the practice of exergames would have acute effects on mood and self-esteem of children during school physical education classes and, in addition, to compare the effects of the practice of exergames with routine curricular physical education classes. Thus, our study is based on two hypotheses. First, there are differences between boys and girls regarding moods and self-esteem when practicing exergames in school physical education compared to that in traditional classes. Another hypothesis is that there are differences between boys and girls.

## Materials and methods

This is a cluster-randomized natural experimental study and controlled with two groups [40,43] comparing the acute effects of exergame practice and routine curricular physical education classes.

### Participants

All participants belonged to a private elementary school localized in urban and central part of Florianópolis, Santa Catarina, Brazil. The school was selected for having qualified teachers and for being one of the largest and most structured schools with ideal facilities for applying exergames and for offering twice a week physical education classes lasting 40 minutes each. Regarding socioeconomic status, approximately 75% of children were from families with medium to high socioeconomic status and 25% were scholarship students from lower social classes.

This study is innovative because it is characterized as a natural experiment [40,43–45] and with high ecological validity [28,38,46]. An experimental design with parallel groups was followed in this study.

In the school investigated, the adopted teaching system is subdivided by class, in which case, the randomization by subjects became unfeasible. Cluster design was a strategy used to minimize confounding effects and other biases in the educational context, so class randomization was performed [47].

This study was conducted during the school term (May of 2018) with children from 10 different classes (clusters). The number of students per class ranged from 20 to 25 students.

A sample of 213 children aged 7–11 years were invited to participate in the study. Inclusion criteria were: (a) to be regularly enrolled in school; (b) not have any physical, cognitive, or mental disability, according to the student's existing records at school; and (c) have no commitment that would prevent him from responding to the instruments. Of the total, 15 children were not present on the day of study presentation, four did not meet the participants' inclusion criteria, and seven were excluded for other reasons. Overall, there was an effective participation rate of 65.73% of eligible children who performed all the procedures required in this experiment. When analyzed between groups, the participation rate was greater than 70% (Fig 1). In terms of percentage, the participation in our study can be considered moderate to high compared with other studies with the theme of exergames in the school context, as the participation rates of similarly themed published studies have varied between 18 and 97% [48]. This reduction in participation rate was due to the absence of students due to illness, travel, or late arrivals during the sessions. In research conducted in educational contexts, this sample loss is

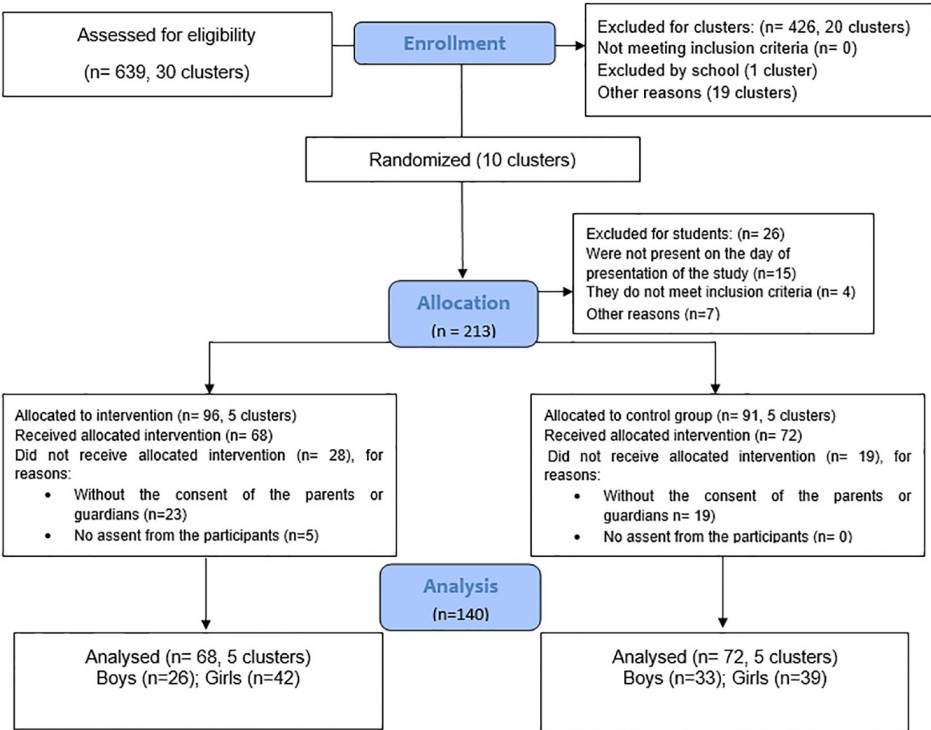

**Fig 1. CONSORT flow chart of clusters and participants within the study (Group intervention—EG and control group—PE group).** EG, experimental group; PE, physical education.

considered acceptable due to school routines. Despite all the peculiarities related to sample loss and participation rate, considering the school context is essential for the success of a physical activity program, whether long or short term [10,48].

Finally, 187 children were allocated to the exergames (EG; n = 96) and control (PE group; n = 91) groups. Of these, 42 were not authorized by their parents or guardians and five did not agree to participate in the interventions. Thus, 68 children (26 boys and 42 girls) in the EG group and 72 children (33 boys and 39 girls) in the PE group were part of the final analysis, totaling 140 children (59 boys and 81 girls, average age 9.41 ± 0.48 years) (Fig 1).

All children who were not included in the study participated in the physical education classes programmed by the school or performed the practice of exergames as pedagogical content after the study ended.

A sample calculation was performed to determine the minimum sample size of participants. A power analysis [49] indicated a minimum of 112 participants are required to compose the appropriate total sample ($\alpha = 0.05$ and $\beta = 0.95$) to identify differences between groups and outcomes. In this controlled experiment, our sample exceeded by more than 20% the number required to perform the appropriate statistical procedures.

## Data collection procedures

The study was approved by the Human Research Ethics Committee of the State University of Santa Catarina (Opinion No. 2.690.067/2018). Before data collection, a pilot study was carried out with ten children in the age group of the present study, to verify the level of understanding of the instruments by the participants and to verify that the children would not respond randomly. In a second step, the children who participated in the study received detailed instructions

on how to complete the instruments. In addition, during the time designated for completion of the instruments, two researchers remained in the classroom to clarify issues and fill any gaps in the procedural recall and understanding needed for proper completion of the instruments. We identified in this pilot study that the children were able to concentrate without any distraction that would hinder the progress of the research.

All children who agreed to participate in the study also obtained parental or guardian consent by the date of the first collection. After agreeing to participate in the study and submitting the consent form with parent/guardian consent, the children performed the first assessment. This initial assessment was called baseline, consisting of instruments related to demographic, anthropometric characteristics, mood, and self-esteem.

All procedures for both data collection and intervention took place at the school where the interventions were performed. The children were evaluated in rooms equipped with chairs and tables and performed individually without interference from any other participant. All children received the support of responsible researchers while filling out the instruments.

The EG performed exergame sessions during physical education classes, and the acute effect was evaluated immediately after the third session (totaling 120 minutes). The physical education group performed its routine curricular physical education classes proposed by the school (totaling 120 minutes), and the acute effect was evaluated immediately after the third class. For both groups, interventions occurred on days apart (e.g., first session: Monday; second session: Wednesday; and third session: Monday again or first session: Tuesday; second session Thursday and third session: Tuesday again).

## Experimental protocol

**Exergame sessions.** In this natural experiment, both the application of the experimental protocol and the routine physical education classes took place in the reality of classes at school. This protocol was similar to another study involving the practice of exergames and conducted in a school environment [16]. A trained teacher taught classes with exergames for the EG. The classroom was equipped with a Kinect® XBOX console (Microsoft, Redmond, WA USA), which allowed up to four students to play simultaneously (Score Subgroup) and each subgroup commanded the game at least twice during the session; the applied game was Just Dance 2015® (Ubisoft Entertainment, SA, Montreuil, France). In addition, this game enabled all students involved (NO Score Subgroup) to participate actively without, however, being in control of the Kinect® functions. Each session lasted 40 minutes and was subdivided into four steps consisting of: (i) brief guidance to familiarize participants with session procedures of the exergames. These instructions were standardized for all classes involved in the study; (ii) activation and seeding with random music available in the game; (iii) performing the required tasks and rotating the subgroups as the song ends; and (iv) cooling down.

**Physical education classes.** The physical education classes have targeted the development of basic motor skills and physical activity using play, games, and sports. During classes, the teachers made adaptations in their traditional teaching approach. These adaptations were mixed with cooperative and competitive activities that simulate sports with playful approaches, for example, shooting at a distant target, simulating basketball, handball, or soccer. All activities offered in physical education classes were planned considering the age, ability level, and previous ability of the students.

Furthermore, consisted of regular activities previously chosen by the class teachers without any interference from the researchers. The classes lasted 40 minutes and were subdivided into four steps: (i) warm-up, (ii) development of technical physical skills, (iii) sports games adapted for children, and (iv) cooling down.

## Measures

**Sociodemographic and anthropometric characteristics.** Demographic information (age and sex) was obtained using a questionnaire and was answered by the children themselves. At the time of the evaluations, teachers and researchers were present to guide the participants. Anthropometric data were verified by body mass (in kg) and height (in meters) using a digital scale with stadiometer (Model Filizola PL 200 kg [Goiânia, Brazil], calibrated and certified by The National Institute of Metrology, Standardization and Industrial Quality [INMETRO], Brazil). Each child was measured and weighed in an appropriate place in the physical education room.

**Brunel mood scale.** The instrument used to assess mood was the Brunel Mood Scale. Studies with children used this instrument to verify the effect of exergames on children in the school environment [16, 21]. The instrument is a Likert-type scale composed of 24 items with five response options, ranging from nothing (0), slightly (1), moderately (2), quite (3), to extremely (4). The participant chooses the option that best fits at that moment. The Brunel Mood Scale evaluates six mood dimensions, classified into psychological (mood depression, anger, and mental confusion) and psychosomatic (fatigue, tension, and vigor) (Table 1). The score of each dimension ranges from 0 to 16 points, where the higher the score, the larger the dimension. A pilot study was performed to identify Cronbach's $\alpha$ (alpha) value. Considering our population (i.e., Brazilian children in a school environment), the result revealed that BRUMS is a reliable instrument to measure the six domains of mood state in children ($\alpha$ = 0.781) [50].

**Rosenberg self-esteem scale.** The instrument used in this study was the Rosenberg Self-Esteem Scale [51]. This is a one-dimensional measure consisting of ten statements related to a set of feelings of self-esteem and self-acceptance that assess overall self-esteem. The measure of self-esteem allows a total score ranging from 10 to 40, increasing in function of the level of self-esteem. Its definition is described in Table 1. All items were submitted to the process of cross-cultural adaptation and validation, and in this study the version adapted for the Portuguese language was used [52]. Isomaa et al. (2013) suggest a cutoff of 25 for classifying low self-esteem in girls and boys [53]. Items are answered on a four-point Likert scale ranging from strongly agree, agree, disagree, and strongly disagree.

## Statistical analysis

Descriptive statistics (mean, standard deviation, and frequencies) were used to identify the general characteristics of the participants. Data normality was verified by the Kolmogorov-

**Table 1. Variables and definitions***.

| Variables | Definition |
|---|---|
| Tension | State of musculoskeletal tension and worry |
| Depression | Emotional state of despondency, sadness, and unhappiness |
| Anger | State of hostility towards others |
| Vigor | Sate of energy and physical stamina |
| Fatigue | State of tiredness or low energy |
| Mental confusion | State of light-headedness and instability |
| Self-esteem | A feeling of self-appreciation |

*Source: Brandt et al. [37], Brandt, Bevilacqua, Andrade [38]. Adaptations are themselves works protected by copyright.

Smirnov test. First, an exploratory analysis was performed comparing the sexes (boys vs. boys, girls vs. girls, and boys vs. girls) and conditions (EG exergames vs. PE classes). For intragroup comparisons (pre- and post-acute effect) the Wilcoxon signed-rank test was used and for inter-group comparisons (sex and condition) the Mann-Whitney *U* test was used.

A two-way multivariate analysis of variance (MANOVA) was used for verifying the differences in mood states and self-esteem by sex and group factors. When the MANOVA detected significant differences, two two-way repeated measures ANOVAs with post-hoc pairwise comparisons were performed. One for comparisons between the groups (EG exergames and PE classes) within each sex (boys and girls). Therefore, the dependent variables were the six domains of mood state (anger, fatigue, mental confusion, depression, vigor, and tension) and self-esteem. Time (pre- and post-intervention) was considered a within-subject factor and group (EG exergames and PE classes) was considered a between-subject factor. The second two-way repeated measures ANOVA corresponds to comparisons between sexes (boys and girls) within each group (EG exergames and PE classes). In this case, the dependent variables were the six domains of mood state (anger, fatigue, mental confusion, depression, vigor, and tension) and self-esteem. Time (pre- and post-intervention) was considered a within-subject factor and sex (boys and girls) was considered a between-subject factor.

To verify the magnitude of intra- and intergroup changes, the effect size proposed by Cohen (1988) was calculated. Considering the criteria, Cohen's *d* values were between 0.2 and 0.5 = small effect, between 0.5 and 0.8 = moderate effect, and $\geq 0.8$ = large effect [54].

All statistical procedures were performed using SPSS software (version 20.0; IBM Corp., Armonk, NY USA). The significance level adopted was p <0.05.

## Results

Regarding internet access in terms of daily hours, the average was higher in boys than in girls from the PE group (p<0.05). In both groups, the average number of hours of dedication to video games per day was higher for boys than for girls (p<0.05) (Table 2). Regarding sports practice across the sample (n = 140), most children (87.1%) played sports in addition to PE classes. Among the boys in the EG group, 96.2% played sports in addition to PE, while in the PE group 87.9% played. Among girls, 84% of the EG group and 87.2% of the PE group engaged in activities other than PE.

Table 3 presents the results (mean and standard deviation [SD]) of the six domains of mood states and the self-esteem scores of the respective groups (EG exergames and PE classes) subdivided by sex (boys and girls), pre- and post-intervention. The definitions of each variable

**Table 2. Description of the anthropometric characteristics of the sample (EG and PE group) and comparison regarding the use of video games and internet access (hours per day) between boys and girls.**

| Variables | Whole Sample (n = 140) | EG (n = 68) | | | PE group (n = 72) | | |
|---|---|---|---|---|---|---|---|
| | | Boys (n = 26) | Girls (n = 42) | p value | Boys (n = 33) | Girls (n = 39) | p value |
| Age (Years) | 9.41 (0.48) | 8.77 (0.58) | 8.88 (0.39) | 0.44 | 9.46 (0.77) | 9.38 (0.62) | 0.30 |
| Weight (kg) | 35.40 (8.30) | 33.39 (7.54) | 33.82 (7.52) | 0.65 | 38.81(9.06) | 35.31 (8.23) | 0.11 |
| Height (m) | 1.38 (0.67) | 1.34 (0.62) | 1.35 (0.50) | 0.93 | 1,42 (0.55) | 1.39 (0.71) | **0.04** |
| BMI (kg/m$^2$) | 18.37 (3.14) | 18.21 (3.06) | 18.38 (2.99) | 0.55 | 19.00 (3.81) | 17.92(2.72) | **0.31** |
| V.G (h/day) | 0.90 (1.80) | 2.30 (2.90) | 0.36 (0.66) | **<0.01** | 1.31 (2.07) | 0.29 (0.68) | **<0.01** |
| Int. A. (h/day) | 1.99 (2.21) | 3.14 (3.80) | 1.43 (1.30) | 0.06 | 2.54 (2.25) | 1.52 (1.29) | **0.01** |

*Abbreviations*: EG = experimental group; PE = physical education; BMI = body mass index; V.G = Video game; Int. A. = Internet access.

**Table 3. Description (mean and standard deviation [SD]) of the six domains of mood states and self-esteem.**

| Variables | Exergames Mean (SD) | | | | PE classes Mean (SD) | | | |
|---|---|---|---|---|---|---|---|---|
| | Boys (n = 26) | | Girls (n = 42) | | Boys (n = 33) | | Girls (n = 39) | |
| | Pre | Post | Pre | Post | Pre | Post | Pre | Post |
| Mood States | | | | | | | | |
| Tension | 3.73 (2.55) | 2.42 (2.35) | 3.00 (2.67) | 1.76 (2.01) | 2.30 (1.99) | 1.45 (1.97) | 2.77 (2.42) | 1.64 (1.59) |
| Depression | 0.42 (0.94) | 0.65 (1.81) | 0.53 (1.97) | 0.26 (0.76) | 0.48 (1.12) | 0.12 (0.41) | 0.46 (0.96) | 0.21 (0.46) |
| Anger | 0.88 (1.68) | 1.23 (2.40) | 0.42 (1.75) | 0.21 (0.97) | 0.70 (1.89) | 0.24 (0.96) | 0.62 (1.29) | 0.23 (0.58) |
| Vigor | 9.73 (3.35) | 10.73 (2.89) | 9.77 (3.27) | 10.21 (3.23) | 9.18 (2.87) | 9.33 (3.12) | 9.18 (2.66) | 8.46 (2.98) |
| Fatigue | 1.62 (2.38) | 1.42 (1.74) | 0.93 (2.23) | 1.69 (2.85) | 1.94 (2.93) | 0.73 (1.84) | 1.85 (2.21) | 1.31 (2.19) |
| Mental confusion | 1.19 (1.41) | 0.88 (1.36) | 1.43 (2.03) | 0.74 (1.86) | 1.06 (1.39) | 0.55 (1.60) | 1.95 (1.91) | 1.08 (1.42) |
| Self-esteem | | | | | | | | |
| Score | 32.38 (3.92) | 32.73 (3.53) | 31.88 (4.53) | 33.45 (4.29) | 32.55 (3.50) | 34.58 (4.46) | 32.33 (3.32) | 33.74 (4.29) |

are presented in Table 1. There were only significant group effects (two-way MANOVA: Wilks' λ (lambda) = 0.85; F[7,130] = 3.20; $p$ = 0.004; $\eta^2$ = 0.15) on mood states and self-esteem.

### Analysis of mood states and self-esteem in boys

A repeated measures ANOVA demonstrated time effects on mental confusion (F[1,57] = 5.51, p = 0.02), tension (F[1,57] = 17.37, p <0.01), and self-esteem (F[1,57] = 6.40; p = 0.01) (Table 4). After three sessions, the tension presented significant differences between the groups (F[1,57] = 5.38; p = 0.02), with tension being higher among boys who practiced exergames compared to the PE group (mean ± SD: 2.42 ± 2.35 vs. 1.45 ± 1.97).

### Analysis of mood states and self-esteem in girls

Among girls, the effect of time on mental confusion (F[1,77] = 13.01; p <0.01), tension (F[1,77] = 19.67; p <0.01), and self-esteem score (F[1,79] = 20.59; p <0.01) was verified (Table 5). The repeated measures ANOVA identified interactions between time and groups for fatigue (F[1,77] = 5.26; p = 0.02). After three sessions, girls from the EG showed superior results for vigor than those from the PE group (mean ± SD: 10.21 ± 3.23 vs. 8.46 ± 2.98; F[1,77] = 5.46; p = 0.02).

**Table 4. Analysis of variance of repeated measures on mood states and self-esteem among boys.**

| | Vigor | | Depression | | Anger | | Fatigue | | Mental confusion | | Tension | | Self-esteem | |
|---|---|---|---|---|---|---|---|---|---|---|---|---|---|---|
| | F | p | F | p | F | p | F | p | F | p | F | p | F | p |
| Time | 1.91 | 0.17 | 0.11 | 0.74 | 0.04 | 0.84 | 3.84 | 0.05 | 5.51 | **0.02** | 17.37 | **<0.01** | 6.40 | **0.01** |
| Time x playing EG | 1.03 | 0.31 | 2.18 | 0.14 | 2.21 | 0.14 | 2.02 | 0.16 | 0.35 | 0.55 | 0.78 | 0.37 | 3.21 | 0.07 |
| Exergames x PE group | 1.93 | 0.17 | 1.11 | 0.29 | 2.38 | 0.12 | 0.17 | 0.70 | 0.48 | 0.48 | 5.38 | **0.02** | 1.50 | 0.22 |

**Table 5. Repeated measures analysis of variance on mood and self-esteem in girls.**

| | Vigor | | Depression | | Anger | | Fatigue | | Mental confusion | | Tension | | Self-esteem | |
|---|---|---|---|---|---|---|---|---|---|---|---|---|---|---|
| | F | p | F | P | F | p | F | p | F | p | F | p | F | p |
| Time | 0.01 | 0.93 | 2.84 | 0.96 | 3.77 | 0.05 | 0.11 | 0.73 | 13.01 | **<0.01** | 19.67 | **<0.01** | 20.59 | **<0.01** |
| Time x playing EG | 2.84 | 0.09 | 0.01 | 0.89 | 0.37 | 0.54 | 5.26 | **0.02** | 0.15 | 0.69 | 0.08 | 0.92 | 0.06 | 0.80 |
| Exergames x PE group | 5.46 | **0.02** | 0.04 | 0.84 | 0.17 | 0.67 | 0.39 | 0.53 | 1.56 | 0.21 | 0.23 | 0.63 | 0.18 | 0.66 |

**Table 6. Repeated measures analysis of variance on mood states and self-esteem in the EG group.**

|  | Vigor | | Depression | | Anger | | Fatigue | | Mental confusion | | Tension | | Self-esteem | |
|---|---|---|---|---|---|---|---|---|---|---|---|---|---|---|
|  | F | p | F | P | F | p | F | p | F | p | F | p | F | p |
| Time | 3.19 | 0.08 | 0.02 | 0.89 | 0.08 | 0.78 | 0.65 | 0.42 | 4.61 | **0.03** | 20.22 | **<0.01** | 4.23 | **0.04** |
| Time x Sex | 0.14 | 0.70 | 1.20 | 0.27 | 1.01 | 0.29 | 1.94 | 0.17 | 0.70 | 0.40 | 0.06 | 0.82 | 1.79 | 0.18 |
| Boys x Girls | 0.03 | 0.84 | 0.33 | 0.56 | 4.57 | **0.03** | 0.21 | 0.65 | <0.01 | 0.92 | 1.51 | 0.22 | 0.13 | 0.71 |

## Analysis of mood states and self-esteem in the EG group

Regarding sex comparisons in the EG exergames group, there was an effect of time on mental confusion ($F[1,64] = 4.61$; $p = 0.03$), tension ($F[1,64] = 20.22$; $p <0.01$), and self-esteem score ($F[1,66] = 4.23$, $p = 0.04$). After three sessions of exergames, boys had higher levels of anger than girls (mean; SD: $1.23 \pm 2.40$ vs $0.21 \pm 0.97$; $F[1,64] = 4.57$; $p = 0.03$) (Table 6).

## Analysis of mood states and self-esteem in the PE group

Repeated measures ANOVA demonstrated time effects on most domains of mood, depression ($F[1,70] = 7.39$; $p <0.01$), anger ($F[1,70] = 8.52$; $p <0.01$), fatigue ($F[1,70] = 8.49$; $p <0.01$), mental confusion ($F[1,70] = 14.27$; $p <0.01$) and tension ($F[1,70] = 15.06$; $p <0.01$) in addition to self-esteem ($F[1,70] = 27.71$; $p <0.01$). After three routine curricular physical education classes, boys had lower scores for mental confusion than girls (mean $\pm$ SD: $0.55 \pm 1.60$ vs. $1.08 \pm 1.42$; $F[1,70] = 4.56$; $p = 0.03$) (Table 7).

## Effect sizes and complementary analysis

In comparisons using the Wilcoxon signed-rank and paired Mann-Whitney $U$ tests, significant differences and effect sizes ranging from small to large were identified (Table 8). The boys presented with lower stress levels, with smaller magnitudes (PE group: $p <0.05$; Cohen's $d = 0.43$) and moderate magnitudes (EG: $p <0.05$; Cohen's $d = 0.51$); lower depression scores with large effect (PE group: $p <0.05$; Cohen's $d = 0.88$); lower fatigue rates with moderate magnitudes (PE group: $p <0.05$; Cohen's $d = 0.66$) and reduced mental confusion with small effect (PE group: $p <0.05$; Cohen's $d = 0.31$). In addition, increased self-esteem with moderate magnitude (PE group: $p <0.05$; Cohen's $d = 0.58$) (Table 8).

The girls presented with stress reduction with moderate magnitude (EG: $p <0.01$; Cohen's $d = 0.62$; PE group: $p <0.01$; Cohen's $d = 0.55$); lower levels of depression with moderate effect (PE group: $p <0.05$; Cohen's $d = 0.71$); lower values for fatigue with small effect (PE group: $p <0.05$; Cohen's $d = 0.24$); in addition to reduced mental confusion with moderate magnitude (PE group: $p <0.05$; Cohen's $d = 0.61$) (Table 8).

In the EG, boys had higher rates of anger and greater effect than girls at the time after intervention (EG: $p <0.05$; Cohen's $d = 0.61$). In the PE group, small effects were identified in the variables fatigue (EG: $p <0.05$; Cohen's $d = 0.28$) and mental confusion (EG: $p <0.05$; Cohen's $d = 0.38$), both being smaller for boys than girls (Table 8).

**Table 7. Repeated measures analysis of variance on mood states and self-esteem in the PE group.**

|  | Vigor | | Depression | | Anger | | Fatigue | | Mental confusion | | Tension | | Self-esteem | |
|---|---|---|---|---|---|---|---|---|---|---|---|---|---|---|
|  | F | p | F | P | F | p | F | p | F | p | F | p | F | p |
| Time | 0.56 | 0.45 | 7.39 | **<0.01** | 8.52 | **<0.01** | 8.49 | **<0.01** | 14.27 | **<0.01** | 15.06 | **<0.01** | 27.71 | **<0.01** |
| Time x sex | 1.32 | 0.25 | 0.22 | 0.64 | 0.05 | 0.80 | 1.25 | 0.26 | 0.94 | 0.33 | 0.30 | 0.58 | 0.90 | 0.34 |
| Boys x Girls | 057 | 0.45 | 0.03 | 0.84 | 0.03 | 0.85 | 0.28 | 0.59 | 4.56 | **0.03** | 0.65 | 0.42 | 0.41 | 0.52 |

**Table 8. Summary of effect sizes (ES) and 95% confidence intervals (95% CIs) of the investigated sample (n = 140) subdivided into condition (EG and PE group) and sex (boys and girls) and comparison of intergroup effects after intervention with EG and PE group.**

| Variables | EG Effect Size (CI 95%) | | | PE Group Effect size (CI 95%) | | | EG x PE Group Effect Size (CI 95%) | |
|---|---|---|---|---|---|---|---|---|
| | Boys | Girls | Boys x Girls | Boys | Girls | Boys x Girls | Boys x Boys | Girls x Girls |
| | Pre x post | Pre x post | Post x post | Pre x post | Pre x post | Post x post | Post x post | Post x post |
| Mood States | | | | | | | | |
| Tension | **0.51**[a] | **0.62**[a] | 0.30 | **0.43**[a] | **0.55**[a] | 0.11 | 0.45 | 0.07 |
| | **(-0.03–1.0)** | **(0.17–1.05)** | (-0.18–0.80) | **(-0.06–0.92)** | **(-1.19–0.08)** | (-0.36–0.57) | (-0.7–0.97) | (-0.37–0.50) |
| Depression | -0.13 | 0.35 | 0.30 | **0.88**[a] | **0.71**[a] | 0.21 | 0.43 | 0.08 |
| | (-0.67–0.41) | (-0.07–0.78) | (-0.18–0.80) | **(-0.37–1.38)** | **(0.25–1.16)** | (-0.25–0.67) | (-0.09–0.94) | (-0.36–0.51) |
| Anger | -0.15 | 0.21 | **0.61**[b] | 0.47 | 0.67 | 0.13 | **0.56**[b] | 0.02 |
| | (-0.69–0.40) | (-0.21–0.64) | **(0.11–1.11)** | (-0.01–0.96) | (0.21–1.13) | (-0.45–0.48) | **(-0.43–1.09)** | (-0.41–0.46) |
| Vigor | 0.30 | 0.13 | 0.16 | 0.05 | -0.27 | 0.29 | 0.46 | **0.56**[b] |
| | (-0.25–0.84) | (-0.29–0.56) | (-0.32–0.65) | (-0.42–0.53) | (-0-72–0.17) | (-0.18–0.75) | (-0.06–0.98) | **(0.12–1.01)** |
| Fatigue | 0.11 | -0.26 | 0.10 | **0.66**[a] | **0.24**[a] | **0.28**[b] | 0.38 | 0.15 |
| | (-0.42–0.65) | (-0.69–0.16) | (-0.38–0.60) | **(0.16–1.15)** | **(-0.19–0.69)** | **(-0.18–0.75)** | (-0.13–0.90) | (-0.28–0.58) |
| Mental confusion | 0.22 | **0.37**[a] | 0.08 | **0.31**[a] | **0.61**[a] | **0.35**[b] | **0.35**[b] | **0.35**[b] |
| | (-0.31–0.77) | **(-0.06–0.80)** | (-0.40–0.57) | **(-0.17–0.80)** | **(0.15–1.06)** | **(-0.11–0.82)** | **(-0.11–0.82)** | **(-0.11–0.82)** |
| Self-esteem | | | | | | | | |
| Score | 0.08 | 0.34 | 0.18 | **0.58**[a] | 0.42 | 0.19 | **0.22**[b] | 0.20 |
| | (-0.45–0.63) | (-0.08–0.77) | (-0.32–0.68) | **(0.08–1.07)** | (-0.02–0.87) | (-0.27–0.66) | **(-0.29–0.74)** | (-0.23–0.44) |

[a]. The Wilcoxon test identified a significant difference at the P < 0.05 level in the pre and post-test comparisons between the EG and PE group.

[b]. The Mann-Whitney *U* test identified a significant difference at the P < 0.05 level in the post test comparisons between boys and girls EG group and boys and girls PE group, and difference in the post-test between boys EG and PE group and difference in the post-test between girls EG and PE group.

In the intergroup analyses, boys from the PE group presented lower rates of anger and mental confusion, with moderate and small magnitudes (Boys: p <0.05, Cohen's *d* = 0.56; Boys = p <0.05, Cohen's *d* = 0.35, respectively), than those from the EG. In addition, self-esteem was higher in boys from the PE group, with small magnitude, than in those from the EG (Boys: p <0.01; Cohen's *d* = 0.22) (Table 8). Among girls, the EG had moderate magnitudes and higher results for vigor (Girls: p <0.01; Cohen's *d* = 0.56) than the PE group. Mental confusion was lower for the EG, and showed little effect, than the PE group (Girls: p <0.05; Cohen's *d* = 0.35) (Table 8).

## Discussion

The objective of this study was to determine if the practice of exergames has an acute effect on mood and self-esteem of boys and girls during school physical education classes and to compare the effects of the practice of exergames with routine curricular physical education classes. This natural experiment showed high ecological validity since there was no interference in the usual routine of the students [43]. Robertson et al. [10] suggest that considering the environment and context contributes to more robust results, especially when the intervention performed does not change the routine in the school environment. Innovation in physical education classes was characterized by including technological tools as experimentation in the participants' natural context [10,44,45].

In our study, the positive acute effect on the psychological health of the students was verified and this supported the inclusion of little explored pedagogical resources into the physical education classes. Our study also demonstrated the possibility of the association between

technology and body movement culture within schools, without, however, replacing traditionally enjoyable practices such as sports and games [55]. Studies conducted in the school environment have identified that the practice of exergames can promote increased physical activity and improved mental health and may be a potential educational tool [21,28,32,56,57]. Nevertheless, there are few studies that have investigated the practice of exergames as part of school physical education and its real effectiveness, requiring further study [58,59].

Our results reveal that the conditions (exergames and PE classes) and the sexes (boys and girls) presented differences. Overall, the average hours per day playing video games considered sedentary was higher for boys than girls, corroborating other studies [6,60]. In recent years there has been increasing concern regarding the excessive use of video games and the Internet due to the association with psychological and social comorbidities such as depression, attention deficit disorder, hyperactivity, alcohol intake, anxiety, and low psychosocial support [6,12]. For children, the fundamental role of physical education at school in raising students' awareness of the importance of healthy practices in guiding the substitution of harmful practices is highlighted [21,34].

## Mood

Regarding the mood of children in this study, our overall results revealed that the practice of exergames compared to traditional physical education classes produced similar effects, presenting higher levels of vigor while the other domains (tension, depression, anger, fatigue, and mental confusion) remained low, indicating good mental health [61]. Our results demonstrate that acute effects of the practice of exergames and PE classes were positive for both boys and girls. Lee et al. (2017) investigated mood states in children during PE classes and identified results similar to ours [21]. In another study, Gao et al. [28] proposed that the practice of exergames and physical education classes produce similar effects when analyzed in the context of different levels of intensity (light, moderate and vigorous), energy expenditure, and sedentary behavior [28]. Our study did not evaluate the intensity of the exercise sessions, however, considering the participants' age, they were instructed to modulate the intensity according to their personal choice. The effect of exergames during physical education classes demonstrated success in children with light and moderate intensity [62].

In the pre-intervention moment, tension was high for boys and girls in both groups. Tension can be characterized by feelings such as nervousness, apprehension, worry, and anxiety [63]. High tension has revealed that this domain in mood is sensitive before physical education classes for children. In our study, tension decreased significantly between boys and girls after three sessions in both groups. This reduction considering the conditions and the participant's sex can be explained due to the adaptations that the activities generated in the children. The novelty of the activity that diminished over time generated a natural relaxation in the students, so they began to dominate the components of the class session. Activities that offer greater control of motor actions such as breathing and postural control tend to be more effective in the long run [41].

In an isolated analysis of the exergames group, it is important to highlight that there are differences between playing video games individually and collectively, as they have different goals and are still unknown when considering individual preferences, such as age, sex, and level of aptitude [64]. Peng and Hsieh (2012) suggest that playing with friends (multiplayer) can have different impacts on players regarding motivation and engagement to practice, unlike when practiced individually [65]. These factors may also have contributed to reducing the tension generated by success or failure in game tasks [66].

Another finding from our study was that boys' anger was higher than girls' after exergames, while anger levels among boys and girls in PE classes remained low. Anger is characterized as a component that can influence a given task due to increased irritability and frustration [67].

In the case of boys, it was observed that, in part, the preference to participate in traditional sports activities (e.g., soccer) was manifested during the sessions in verbal reports to researchers. In observational analysis of the sessions, the boys showed greater resistance and did not fully accept the proposed practice, which may explain the higher levels of anger. In contrast, the girls showed a decrease in this domain and greater acceptance during the sessions. Our results partially contradict the findings of Lee, Xiang, and Gao [21] who identified, after 30-minute sessions with exergames, decreased anger. However, this study did not consider the influence of the child's sex on their perception of mood of who participated in the study.

After the three sessions of both conditions and sexes, girls from both the EG and PE groups showed a reduction in mental confusion, while only boys from PE group showed a reduction in this domain. In the PE group, results of girls were significantly higher than those of boys. Although this presented with negative mood reduction, it is not possible to conclude if this reduction was associated with sex. It is well known that both the practice of exergames and physical education can bring psychosocial and cognitive impacts, as well as increase social interaction, motivation, attention, and visual-spatial skills [26,32]. With less competitive characteristics and no body contact between the participants, the practice of exergames can be used in an environment that seeks to eliminate possible conflicts between students. Positive relationship between physical activity and mood during physical education classes for children has been little reported in studies [16,21].

## Self-esteem

The average self-esteem score presented in our study, according to the instrument proposed by Rosenberg (1965), was greater than 30, indicating good self-esteem [53]. Children with low self-esteem are at risk of developing psychological and social problems detrimental to their development [68]. Exercise and exergames have positive effects on various aspects of self-esteem in healthy children and adolescents [32,69]. High self-esteem associated with other healthy habits can predict adolescence with greater chances of success in motor and cognitive tasks, and low self-esteem is associated with disorders such as social phobia and depression in this age group [31].

After three sessions, our findings indicated a significant increase in self-esteem in girls in both groups, EG and PE, while only boys in the PE group showed a significant increase. During the practice of exergames the game applied in the experiment was Just Dance and this may have contributed to raising the girls' self-esteem, since this type of game involves the dance content and consequently may have influenced more girls during body practices. The EG showed higher self-esteem than the PE group at the post-intervention moment. Duman et al. (2016) analyzed children who were considered obese and indicated a significant improvement in self-esteem [33]. In another study with children who were considered obese there was a significant increase in self-esteem using a cooperative exergame with both sexes [57], however, the authors identified significant improvement with competitive games. This improvement was greater than our study, however, children and adolescents that are considered obese usually have problems with low self-esteem and body satisfaction [33,34,58,70]. Nevertheless, our study found that a few sessions of exergames are sufficient to increase the self-esteem of healthy children. Performing innovative tasks in the school environment makes the students perceive of the school as a positive space for their self-esteem and cultural identity.

## Innovations, strength of study, practical applications, and limitations

Our study presents innovative aspects. Exergames are easily accessible and involve relatively low costs, with the possibility of adoption by schools due to the low cost. Currently, there is

an online platform (www.justdancenow.com) that, with a single annual subscription, allows you to play and exercise at the same time. It is a natural experiment that provided high ecological validity and proved its effectiveness as a form of physical activity in class. It is a cluster-randomized controlled experiment that included a significant number of participants, and it innovated by transferring theoretical and methodological scientific knowledge into the real school environment reaching the entire student community with an attractive practical proposal.

Annually, schools have made efforts to promote innovative practices used in teaching. Our study offers an opportunity for schools, managers, and teachers to discuss other innovative technologies that can integrate with their practice in the field for promoting physical education, along with sustainable and promising skills. By incorporating innovative technologies and practices, schools can benefit, even in the short term. Our study investigated the effect of exergames in the school environment, in addition to showing positive results for the participants, it demonstrated no harm or negative effects beyond the participation rates compared with other studies. Although conducted in just one school, our study demonstrated feasibility, in addition to the potential for replicability in other school environments and other age groups without major investment, sophisticated spaces, or significant resources.

Our results are encouraging, as we identified and analyzed the emotional responses of students during physical education classes using innovative interventions with exergames. Our study provides important information for the context and educational process. This study compared few exercise sessions with routine curricular physical education. The results showed that exergames provide an opportunity for children to be active in their educational environment using technological resources for physical and psychological health. Active games can improve the mental health of and have no undesirable effects on students.

Children engaged in active video games can benefit from long-lasting effects as well as lifelong active behavior. When considering the significant amount of time spent with traditional video games, especially in childhood, one can replace this passive practice with an active video game for better physical and psychological responses in the future. In addition, the use of exergames can diversify curricular physical education classes, making them more attractive to students. Being an approach that combines technology and body movement is interesting to promote an organized and fun environment attracting the largest number of participants.

In our study, no blinding was performed in the procedures. Future trials may wish to consider this factor to minimize confounding effects during the experiment. The use of only one school to conduct the study makes it difficult to generalize the results. Future trials need to consider increasing the number of schools participating. It is necessary to investigate the long-term effects of exergames compared to routine PE classes and whether they are capable of providing complementary physical activity that produces positive effects on mood, self-esteem, and other variables related to children's mental health (e.g., depression). In this cluster-randomized and parallel group controlled experiment, only one set of exergames was used. Other games with consoles and diverse technology are required (e.g., challenging, sporting, competitive, and cooperative games). Regarding the sample, the lack of information on socioeconomic status, ethnic profile, academic performance, can be considered a limiting factor in our study. Such variables can contribute to a better understanding of the results identified. In addition, the impossibility of applying research instruments individually to participating children should be considered in future studies. Another point refers to the division of groups and overcoming the limitation of parallel groups, individuals can be compared with themselves using a crossover design after an adequate washout period. Our study did not test the intensity of movements performed in either the EG or PE group. This type of testing can contribute to investigate other outcomes that are still little explored when investigating the effects related to

mental health and may contribute to the use of active video games as a complement in traditional curricular physical education classes.

## Conclusions

The results indicate that a few sessions with exergames caused boys and girls to present positive variations in mood and self-esteem. The exergames practice showed higher positive acute effects in girls. However, PE classes were more effective in both boys and girls after three classes. The results indicate the importance of creating a task-oriented environment during the classes, which will promote positive experiences for both sexes, enabling the development of students and encouraging more active behaviors to improve children's mental health.

## Acknowledgments

The authors gratefully acknowledge all the participating students and their parents, without whom the present study could not have been carried out. Additionally, the authors acknowledge all the members of the school for their enthusiasm and collaboration, especially the school principal and the physical education teachers.

## Author Contributions

**Conceptualization:** Whyllerton Mayron da Cruz, Clara Knierim Correia.

**Formal analysis:** Alexandro Andrade, Whyllerton Mayron da Cruz, Clara Knierim Correia.

**Funding acquisition:** Alexandro Andrade.

**Investigation:** Alexandro Andrade, Whyllerton Mayron da Cruz, Clara Knierim Correia, Ana Luiza Goya Santos, Guilherme Guimarães Bevilacqua.

**Methodology:** Alexandro Andrade, Whyllerton Mayron da Cruz, Clara Knierim Correia, Ana Luiza Goya Santos, Guilherme Guimarães Bevilacqua.

**Project administration:** Alexandro Andrade.

**Resources:** Alexandro Andrade.

**Supervision:** Alexandro Andrade, Whyllerton Mayron da Cruz.

**Validation:** Alexandro Andrade, Whyllerton Mayron da Cruz.

**Visualization:** Alexandro Andrade.

**Writing – original draft:** Alexandro Andrade, Ana Luiza Goya Santos, Guilherme Guimarães Bevilacqua.

**Writing – review & editing:** Alexandro Andrade, Whyllerton Mayron da Cruz, Clara Knierim Correia, Ana Luiza Goya Santos, Guilherme Guimarães Bevilacqua.

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
