## [Decision Letter · Decision Letter 0]

7 Jan 2020

PONE-D-19-24839

Effect of practice exergames on the mood states and self-esteem of boys and girls during physical education classes: a cluster-randomized controlled natural experiment

PLOS ONE

Dear Dr Alexandro Andrade,

Thank you for submitting your manuscript to PLOS ONE. After careful consideration, we feel that it has merit but does not fully meet PLOS ONE’s publication criteria as it currently stands. Therefore, we invite you to submit a revised version of the manuscript that addresses the points raised during the review process.

We would appreciate receiving your revised manuscript by 24/2/20. To enhance the reproducibility of your results, we recommend that if applicable you deposit your laboratory protocols in protocols.io, where a protocol can be assigned its own identifier (DOI) such that it can be cited independently in the future. For instructions see: http://journals.plos.org/plosone/s/submission-guidelines#loc-laboratory-protocols

We look forward to receiving your revised manuscript.

Kind regards,

Eduardo Fonseca-Pedrero, PhD

Academic Editor

PLOS ONE

Additional Editor Comments:

The work entitled “Effect of practice exergames on the mood states and self-esteem of boys and girls during physical education classes: a cluster-randomized controlled natural experiment” is of great interest in the mental health field. The research is very stimulating. As comments I would like to say:

1.- According to criterion #3: ""Experiments, statistics, and other analyses are performed to a high technical standard and are described in sufficient detail." (https://journals.plos.org/plosone/s/criteria-for-publication#loc-3), Please add: " post-hoc corrections for multiple comparisons " and "effect sizes".

2.- Add the main hypotheses at the end of the intro.

3.-Add more information about the all sociodemographic characteristics of the sample, e.g., ethnicity, socio-economic status, etc.

4.-Do you have any information about non-response? Describe inclusion/exclusion criteria if part of the data was excluded from the analysis. Were outliers removed from the data? Please, Which method did you use to deal with missing data in the analyses? What variables are related to missing data?

5.- Add information about the main goals on each physical education session.

6.- Add psychometric properties of measures used in this study (Omega, validity evidences).

7.- I am wondering if you can assess psychological variables with self-reports tools in this age group.

8.- Please compute a MANOVA intra/inter (report, Wilks’ lambda and eta square) (see also point 1).

9.- Add subheadings in Results section.

10.- Please, add information about the limitations of the present study (e.g., sample, measures, etc.)

11.- Add degrees of freedom.

12.- Please move the First paragraph of the results section to participants section.

13.- I don’t know if is better talk about sex or gender.

14.- Please, follow carefully Plos One guidelines (tables, subheadings, etc.)

15.- You don’t have considered relevant covariables like IQ, school achievement, if boy and girls practise other sports at the same time, etc.

2. Please refer to any post-hoc corrections to correct for multiple comparisons during your statistical analyses. if these were not performed please justify the reasons.

4. We note you have included a table to which you do not refer in the text of your manuscript. Please ensure that you refer to Table 4 in your text; if accepted, production will need this reference to link the reader to the Table.

Reviewers' comments:

Reviewer's Responses to Questions

**Comments to the Author**

1. Is the manuscript technically sound, and do the data support the conclusions?

Reviewer #1: Yes

Reviewer #2: Yes

2. Has the statistical analysis been performed appropriately and rigorously? 

Reviewer #1: Yes

Reviewer #2: Yes

3. Have the authors made all data underlying the findings in their manuscript fully available?

Reviewer #1: No

Reviewer #2: Yes

4. Is the manuscript presented in an intelligible fashion and written in standard English?

Reviewer #1: Yes

Reviewer #2: Yes

5. Review Comments to the Author

Reviewer #1: The authors present a very interesting and methodologically sound work that examines the effects of of practice exergames on the mood states and self-esteem of boys and girls during physical education classes. To this end, the authors designed a cluster-randomized controlled natural experiment with two arms with 96 participants in the experimental group (EG) and 91 participants in the control group (CG). The EG practiced exergames during three 40-minute classes, and the PE group held three routine curricular PE classes.

I consider that the work carried out by the authors is remarkable and very relevant at the social and educational level.

Here are some recommendations to improve the final quality of the manuscript:

1. It would be convenient to include reliability figures of the measuring instruments used, both from the literature and from the data of this study.

2. The percentage of sample lost is 34.27% and the authors indicate in the manuscrit that it is an "acceptable" value with reference to other studies conducted in educational contexts, but the authors do not include any references. It would be convenient to provide sample loss data in other similar studies.

3. I wonder about the assessment process and, in particular, about how the veracity of children's responses to the measuring instruments used was guaranteed. Specifically, I am concerned that some children may have responded randomly to the tests.

4. In line with the previous point, it is necessary to include more information about the children evaluation process: did they respond to the items alone, in group, with study researchers...? How the children understood of the items was guaranteed?...

5.- I have serious concerns about the excessive exposure of technology in schools. I have no doubts about the attractiveness and facilitator effects of these types of programs but I am worried about other potential effects of children overexposure to "screens" at school (smartphones, tablets, laptops, PCs,...). The long-term effects in other psychological variables -for instance on frustration tolerance, emotional regulation strategies or increasing the risk of addiction to new technologies in vulnerable children- of long-term overexposed are unknown. These variables, moreover, have not been measured in this trial, which seems to me to be a significant limitation.

It is clear that the current trend of education is going to incorporate TICS, and children really love them, but I am still worried about its global impact on the psychological development of children. I think it is important to add some reflection on these risks in the manuscript. In addition, it would be convenient to take into account in the manuscript the possibility of developing alternative intervention protocols for children already "over-hooked" to ICTs and/or to reduce this risk in vulnerable childrens and offering alternatives to them.

Reviewer #2: The manuscript “Effect of practice exergames on the mood states and self-esteem of boys and girls during physical education classes: a cluster-randomized controlled natural experiment” is innovative and it offers a new possibility to study an important problem in children, like the psychological health and the sedentary life habits.

Then, I will present the possible adjustments:

1.- It would be recommended add the age of the participants in the title.

2.- Some of the studies you cite use more sessions to evaluate the results. Maybe it would be good to take more time (instead of just 3 sessions) for the validity of the results.

3.- Using only one school to carry out the study makes it difficult to generalize the results. That is, external validity is affected.

4.- How is the evaluation of students participating in the study carried out? Explain the process to select the sample.

5.- The understanding of the questionnaires used may be affected by the age of the participants.

6.- In the "results" section, clarify in the tables what each variable means.

7.- In the "discussion" section, the results are compared with some studies, but in them, the results involve adolescent participants and more time in sessions, such as Simons et al. (2015) or the statements of lines 349 to 353. Maybe the results cannot be compared at the same level.

8.- Did the school have the material used to carry out the exergame?

9.- It would be advisable to add information on the price of the resources used in the study, to study the viability in all schools.

10.- In what month was the study conducted? Had they had physical education sessions already at school?

11.- Were the physical education sessions novel for the studies or otherwise they had already done them?

Add information on these aspects for the replicability of the study.

12.- I recommend adding recent research information:

- “The Psychological Effects of Exergames for Children and Adolescents with Obesity: A Systematic Review and Meta-Analysis” (2019) Andrade, A., Correia, CK., Coimbra, DR. https://www.liebertpub.com/doi/full/10.1089/cyber.2019.0341

6. PLOS authors have the option to publish the peer review history of their article (what does this mean?). If published, this will include your full peer review and any attached files.

Reviewer #1: No

Reviewer #2: No

---

## [Author Response · Author response to Decision Letter 0]

7 Mar 2020

February 24, 2020

Dr. Eduardo Fonseca-Pedrero

Academic Editor

PLOS ONE

Dear Editor, 

The manuscript has been carefully reviewed, and appropriate changes have been made in accordance with the reviewer suggestions. We would like to thank the referees for their comments. They have helped to improve the manuscript quality significantly, and we believe that the manuscript now provides a more balanced and better account of the research. We have used the red color to mark all the revisions in the manuscript.

Below the author's comments on the suggestions of reviewer.

Sincerely,

Alexandro Andrade PhD.

Affiliation: Professor of Santa Catarina State University.

Florianópolis, Santa Catarina, Brazil.

Phone Number: +55 (48) 3664-8677

Email: alexandro.andrade.phd@gmail.com

Additional Editor Comments

General comments:

“The work entitled “Effect of practice exergames on the mood states and self-esteem of boys and girls during physical education classes: a cluster-randomized controlled natural experiment” is of great interest in the mental health field. The research is very stimulating.”

1. According to criterion #3: ""Experiments, statistics, and other analyses are performed to a high technical standard and are described in sufficient detail." (https://journals.plos.org/plosone/s/criteria-for-publication#loc-3), Please add: "post-hoc corrections for multiple comparisons " and "effect sizes".

Author’s comments: We include the post-hoc used in the statistical tests. The item “effect sizes” was careful revised and we included the effect size eta partial squared value on two-way MANOVA test.

2. Add the main hypotheses at the end of the intro.

Author’s comments: We added the main hypotheses at the intro.

3. Add more information about the all sociodemographic characteristics of the sample, e.g., ethnicity, socio-economic status, etc.

Author’s comments: We added data related to the characteristics of the school and the socioeconomic profiles of the students in the manuscript. Our study was carried out in a private non-profit (philanthropic) school. Therefore, our sample was composed of students with different socioeconomic status and ethnicity. We added in the study's limitations the non-inclusion of the specific data of the students because it was not carried out individually with each student.

4. Do you have any information about non-response? Describe inclusion/exclusion criteria if part of the data was excluded from the analysis. Were outliers removed from the data? Please, Which method did you use to deal with missing data in the analyses? What variables are related to missing data?

Author’s comments: When analyzing the instruments filled in by the children, when there was no response or even forgetfulness during filling, the researchers informed individually so that it was filled out as requested. Children who participated in only one of the collections or who, for the reasons described (see flow chart) missed an exergame session, were removed from the sample. We decided to exclude data with missing information due to the analysis and considering the statistical robustness used.

5. Add information about the main goals on each physical education session.

Author’s comments: More information regarding traditional physical education classes was added to the text. Physical education classes aimed to develop basic motor skills and improve physical capabilities.

6. Add psychometric properties of measures used in this study (Omega, validity evidences).

Author’s comments: We included this information: A pilot study was performed to identify Cronbach’s 𝞪 (alpha) value. Considering our population (i.e., Brazilian children in a school environment), the result revealed that BRUMS is a reliable instrument to measure the six domains of mood state in children (𝞪 = 0.781).

 I am wondering if you can assess psychological variables with self-reports tools in this age group.

Author’s comments: In the case of a natural experiment, the instruments applied were validated to be answered by the research participants themselves. In addition, the instruments used have been tested and validated and have been applied in previous studies in children.

7. Please compute a MANOVA intra/inter (report, Wilks’ lambda and eta square) (see also point 1).

Author’s comments: We tested MANOVA two-way and reported the Wilk’s lambda and eta partial squared. 

8. Add subheadings in Results section.

Author’s comments:

Considering this question the following subheading were included: “Analysis of mood states and self-esteem in boys, Analysis of mood states and self-esteem in girls, Analysis of mood states and self-esteem in EG group, Analysis of mood states and self-esteem in PE group”.

9. Please, add information about the limitations of the present study (e.g., sample, measures, etc.)

Author’s comments: The following information was included: “Regarding the sample, the lack of information on socioeconomic status, ethnic profile, academic performance, can be considered a limiting factor in our study. Such variables can contribute to a better understanding of the results identified. In addition, the impossibility of applying research instruments individually to participating children should be considered in future studies. Another point refers to the division of groups and overcoming the limitation of parallel groups, individuals can be compared with themselves using a crossover design after an adequate washout period.”

10. Add degrees of freedom.

Author’s comments: We added degrees of freedom in section Results. 

11. Please move the First paragraph of the results section to participants section.

Author’s comments: It has been done. 

12. I don’t know if is better talk about sex or gender. 

Author’s comments: In a recent study with adolescents published in PLOSONE, the term sex was used [1]. In our study, when investigating children, we considered the use of sex to be more appropriate. Because the English term “gender” can have many different meanings depending on the context, including the nature of the published work and the nationality of the author or reader. We consider the term “sex” to be more accurate and less likely to be misinterpreted.

13. Please, follow carefully Plos One guidelines (tables, subheadings, etc.)

Author’s comments: We carefully revised Plos One guidelines.

14. You don’t have considered relevant covariables like IQ, school achievement, if boy and girls practise other sports at the same time, etc.

Author’s comments: Information about physical activity during the research was included. The other information was included as a limitation.

1. Please refer to any post-hoc corrections to correct for multiple comparisons during your statistical analyses. if these were not performed please justify the reasons.

Author’s comments: We included post-hoc corrections.

2. Please amend your list of authors on the manuscript to ensure that each author is linked to an affiliation. Authors’ affiliations should reflect the institution where the work was done (if authors moved subsequently, you can also list the new affiliation stating “current affiliation ” as necessary).

Author’s comments: The following information was included: 2 Laboratory of Aquatic Biomechanics, Santa Catarina State University, Florianópolis, Brazil.

3. We note you have included a table to which you do not refer in the text of your manuscript. Please ensure that you refer to Table 4 in your text; if accepted, production will need this reference to link the reader to the Table.

Author’s comments: We included the information. 

Review Comments to the Author

Please use the space provided to explain your answers to the questions above. You may also include additional comments for the author, including concerns about dual publication, research ethics, or publication ethics. (Please upload your review as an attachment if it exceeds 20,000 characters).

Reviewer #1: Comments to Author

General comments:

The authors present a very interesting and methodologically sound work that examines the effects of of practice exergames on the mood states and self-esteem of boys and girls during physical education classes. To this end, the authors designed a cluster-randomized controlled natural experiment with two arms with 96 participants in the experimental group (EG) and 91 participants in the control group (CG). The EG practiced exergames during three 40-minute classes, and the PE group held three routine curricular PE classes. I consider that the work carried out by the authors is remarkable and very relevant at the social and educational level.

1. It would be convenient to include reliability figures of the measuring instruments used, both from the literature and from the data of this study.

Author’s comments: We included this information: “A pilot study was performed to identify Cronbach’s 𝞪 (alpha) value. Considering our population (i.e., Brazilian children in a school environment), the result revealed that BRUMS is a reliable instrument to measure the six domains of mood state in children (𝞪 = 0.781).”

2. The percentage of sample lost is 34.27% and the authors indicate in the manuscript that it is an "acceptable" value with reference to other studies conducted in educational contexts, but the authors do not include any references. It would be convenient to provide sample loss data in other similar studies.

Author’s comments: As it is a natural experiment, our study did not predict an early drop-out rate. In a recent systematic review of active video games in schools, Norris et al (2016) presented insufficient details regarding the participation rate in the reviewed studies. Even so, our study showed a participation rate of 65.73% of all eligible students after randomization by cluster (n = 213).

We included this information in manuscript: “Overall, there was an effective participation rate of 65.73% of eligible children who performed all the procedures required in this experiment. When analyzed between groups, the participation rate was greater than 70% (Fig. 1). In terms of percentage, the participation in our study can be considered moderate to high compared with other studies with the theme of exergames in the school context, as the participation rates of similarly themed published studies have varied between 18 and 97% [48].”

3. I wonder about the assessment process and, in particular, about how the veracity of children's responses to the measuring instruments used was guaranteed. Specifically, I am concerned that some children may have responded randomly to the tests.

Author’s comments: We appreciate the reviewer's notes. We included this information in manuscript: “Before data collection, a pilot study was carried out with ten children in the age group of the present study, to verify the level of understanding of the instruments by the participants and to verify that the children would not respond randomly. In a second step, the children who participated in the study received detailed instructions on how to complete the instruments. In addition, during the time designated for completion of the instruments, two researchers remained in the classroom to clarify issues and fill any gaps in the procedural recall and understanding needed for proper completion of the instruments. We identified in this pilot study that the children were able to concentrate without any distraction that would hinder the progress of the research.’

4. In line with the previous point, it is necessary to include more information about the children evaluation process: did they respond to the items alone, in group, with study researchers...? How the children understood of the items was guaranteed?...

Author’s comments: The following information was included: “All procedures for both data collection and intervention took place at the school where the interventions were performed. The children were evaluated in rooms equipped with chairs and tables and performed individually without interference from any other participant. All children were supported by the researchers in completing the instruments.”

5. I have serious concerns about the excessive exposure of technology in schools. I have no doubts about the attractiveness and facilitator effects of these types of programs but I am worried about other potential effects of children overexposure to "screens" at school (smartphones, tablets, laptops, PCs,...). The long-term effects in other psychological variables -for instance on frustration tolerance, emotional regulation strategies or increasing the risk of addiction to new technologies in vulnerable children- of long-term overexposed are unknown. These variables, moreover, have not been measured in this trial, which seems to me to be a significant limitation.

Author’s comments: We agree with the reviewer. Our focus was on acute effect of practice exergames in children. Considering the suggestion of the reviewer, we included in the limitation. 

Additional Reviewer #1 Comment

It is clear that the current trend of education is going to incorporate TICS, and children really love them, but I am still worried about its global impact on the psychological development of children. I think it is important to add some reflection on these risks in the manuscript. In addition, it would be convenient to take into account in the manuscript the possibility of developing alternative intervention protocols for children already "over-hooked" to ICTs and/or to reduce this risk in vulnerable childrens and offering alternatives to them.

Author’s comments: We are grateful for the reviewer's considerations and included in the session practical applications and limitations on risks and damage to physical and mental health caused by passive electronic games and a lack of guidance on overuse. Physical education classes with exergames aim to guide healthy and conscious practices.

Reviewer #2: Comments to Author

General comments:

The manuscript “Effect of practice exergames on the mood states and self-esteem of boys and girls during physical education classes: a cluster-randomized controlled natural experiment” is innovative and it offers a new possibility to study an important problem in children, like the psychological health and the sedentary life habits.

1. It would be recommended add the age of the participants in the title.

Author’s comments: We add elementary school in title: “Effect of practice exergames on the mood states and self-esteem of elementary school boys and girls during physical education classes: a cluster-randomized controlled natural experiment”

2. Some of the studies you cite use more sessions to evaluate the results. Maybe it would be good to take more time (instead of just 3 sessions) for the validity of the results.

Author’s comments: We believe that more sessions may have different results than the one identified in this study. However, the objective was to investigate only three sessions, since analyzing the acute effect on mental health becomes relevant since in schools there is a constant change in the contents planned during curricular physical education. In addition, this understanding of psychological aspects can contribute in the short term to the adequacy of stimulating tasks with a direct impact on students' motivation and engagement.

3. Using only one school to carry out the study makes it difficult to generalize the results. That is, external validity is affected.

Author’s comments: We agree with the reviewer and include it in session Innovations, strength of study, practical applications, and limitations: “The use of only one school to conduct the study makes it difficult to generalize the results. Future trials need to consider increasing the number of schools participating.”.

4. How is the evaluation of students participating in the study carried out? Explain the process to select the sample.

Author’s comments: This study was conducted with children from 10 different classes (clusters). The number of students per class ranged from 20 to 25 students. A sample of 213 children aged 7-11 years were invited to participate in the study. Inclusion criteria were: (a) to be regularly enrolled in school; (b) not have any physical, cognitive, or mental disability, according to the student’s existing records at school; and (c) have no commitment that would prevent him from responding to the instruments. Of the total, 15 children were not present on the day of study presentation, four did not meet the participants’ inclusion criteria, and seven were excluded for other reasons.

5. The understanding of the questionnaires used may be affected by the age of the participants.

Author’s comments: The instruments used were validated and used for this age group, with results published in other studies.

6. In the "results" section, clarify in the tables what each variable means.

Author’s comments: Corrections were made.

7. In the "discussion" section, the results are compared with some studies, but in them, the results involve adolescent participants and more time in sessions, such as Simons et al. (2015) or the statements of lines 349 to 353. Maybe the results cannot be compared at the same level.

Author’s comments: We agree with the reviewer that the comparison does not occur at the same level between children and adolescents and that the exposure time in exegames was not the same. We removed this part of the discussion in the manuscript.

8. Did the school have the material used to carry out the exergame?

Author’s comments: Had no exergames, but was equipped with a screen projector.

9. It would be advisable to add information on the price of the resources used in the study, to study the viability in all schools.

Author’s comments: We included this information in session Innovations, strength of study, practical applications, and limitations “Exergames are easily accessible and involve relatively low costs, with the possibility of adoption by schools due to the low cost. Currently, there is an online platform (www.justdancenow.com) that, with a single annual subscription, allows you to play and exercise at the same time”.

10. In what month was the study conducted? Had they had physical education sessions already at school?

Author’s comments: The school was selected for offering twice a week physical education classes lasting 40 minutes each. We add the following information: During the school term (May of 2018)

11. Were the physical education sessions novel for the studies or otherwise they had already done them? Add information on these aspects for the replicability of the study.

Author’s comments: Consisted of regular activities previously chosen by the class teachers without any interference from the researchers.

12. I recommend adding recent research information:

- “The Psychological Effects of Exergames for Children and Adolescents with Obesity: A Systematic Review and Meta-Analysis” (2019) Andrade, A., Correia, CK., Coimbra, DR. https://www.liebertpub.com/doi/full/10.1089/cyber.2019.0341

Author’s comments: We agree with the reviewer and included the recent research.

1. Silva DAS, Chaput JP, Tremblay MS. Participation frequency in physical education classes and physical activity and sitting time in Brazilian adolescents. PLoS One. 2019;14: 1–14. doi:10.1371/journal.pone.0213785

---

## [Decision Letter · Decision Letter 1]

15 Apr 2020

Effect of practice exergames on the mood states and self-esteem of elementary school boys and girls during physical education classes: a cluster-randomized controlled natural experiment

PONE-D-19-24839R1

Dear Dr. Andrade,

We are pleased to inform you that your manuscript has been judged scientifically suitable for publication and will be formally accepted for publication once it complies with all outstanding technical requirements.

With kind regards,

Eduardo Fonseca-Pedrero, PhD

Academic Editor

PLOS ONE

Additional Editor Comments (optional):

Reviewers' comments:

Reviewer's Responses to Questions

**Comments to the Author**

1. If the authors have adequately addressed your comments raised in a previous round of review and you feel that this manuscript is now acceptable for publication, you may indicate that here to bypass the “Comments to the Author” section, enter your conflict of interest statement in the “Confidential to Editor” section, and submit your "Accept" recommendation.

Reviewer #1: All comments have been addressed

Reviewer #2: All comments have been addressed

2. Is the manuscript technically sound, and do the data support the conclusions?

Reviewer #1: Yes

Reviewer #2: Yes

3. Has the statistical analysis been performed appropriately and rigorously? 

Reviewer #1: Yes

Reviewer #2: Yes

4. Have the authors made all data underlying the findings in their manuscript fully available?

Reviewer #1: Yes

Reviewer #2: Yes

5. Is the manuscript presented in an intelligible fashion and written in standard English?

Reviewer #1: Yes

Reviewer #2: Yes

6. Review Comments to the Author

Reviewer #1: The authors have satisfactorily addressed all the comments made. I have no further comment to make.

Reviewer #2: The comments made have been added and taken into account for the new version. The authors have analyzed the proposals and have modified minimal issues suggested by the reviewer. I consider that the article is therefore suitable for publication.

7. PLOS authors have the option to publish the peer review history of their article (what does this mean?). If published, this will include your full peer review and any attached files.

Reviewer #1: No

Reviewer #2: No

---

## [Editor Report · Acceptance letter]

28 Apr 2020

PONE-D-19-24839R1 

 Effect of practice exergames on the mood states and self-esteem of boys and girls during physical education classes: a cluster-randomized controlled natural experiment 

Dear Dr. Andrade:

I am pleased to inform you that your manuscript has been deemed suitable for publication in PLOS ONE. Congratulations! Your manuscript is now with our production department. 

With kind regards,

on behalf of

Dr. Eduardo Fonseca-Pedrero 

Academic Editor

PLOS ONE